

# Predictability of Arctic Sea Ice Drift in Coupled Climate Models

Simon F. Reifenberg[1,a] and Helge F. Goessling[1]

[1]Alfred Wegener Institute for Polar and Marine Research, Bremerhaven, Germany
[a]now at: MARUM – Center for Marine Environmental Science & Institute of Environmental Physics, University of Bremen, Bremen, Germany

**Correspondence:** Simon Reifenberg (sreifenb@uni-bremen.de)

**Abstract.** Skillful sea ice drift forecasts are crucial for scientific mission planning and marine safety. Wind is the dominant driver of ice motion variability, but more slowly varying components of the climate system, in particular ice thickness and ocean currents, bear the potential to render ice drift more predictable than the wind. In this study, we provide the first assessment of Arctic sea ice drift predictability in four coupled general circulation models (GCMs), using a suite of "perfect-model" ensemble
simulations. We find the position vector from Lagrangian trajectories of virtual buoys to remain predictable for at least 90 (45) days lead time for initializations in January (July), reaching about 80 % of the position uncertainty of a climatological reference forecast. In contrast, the uncertainty of Eulerian drift vector predictions reaches the level of the climatological uncertainty within less than four weeks. Spatial patterns of uncertainty, varying with season and across models, develop in all investigated GCMs. For two models providing near-surface wind data (AWI-CM1 and HadGEM1.2), we find spatial patterns and large
fractions of the variance to be explained by wind vector uncertainty. The latter implies that sea ice drift is only marginally more predictable than wind. Nevertheless, particularly one of the four models (GFDL-CM3) shows a significant correlation of up to $-0.85$ between initial ice thickness and target position uncertainty in large parts of the Arctic. Our results provide a first assessment of the inherent predictability of ice motion in coupled climate models, they can be used to put current real-world forecast skill into perspective, and highlight model diversity of sea ice drift predictability.

## 1 Introduction

More than 120 years have passed between Nansen's empirical "rule of thumb" for sea ice drift (Nansen, 1902) and the latest developments of today's sophisticated dynamic sea ice modeling systems. These ongoing efforts are fuelled by an ever-increasing need for reliable sea ice drift forecasts, as human activity in the Arctic Ocean stepped up considerably. For instance, the distance
sailed by bulk carriers under the Arctic Polar Code increased by 160 % from 2016 to 2019, the number of vessels entering the Arctic grew by 25 % (PAME, 2020), and the amount of sailed kilometers in the Northwest Passage nearly tripled from 1990 to 2015 (Dawson et al., 2018). Furthermore, the scientific MOSAiC expedition relied heavily on ice drift predictions (Nicolaus et al., 2022). At the same time, Arctic sea ice declined in the last decades (Stroeve et al., 2012) and is projected to



retreat further in the near future, eventually leading to a virtually ice-free Arctic Ocean before 2050 (SIMIP Community, 2020).
These prospects bring both new opportunities and risks (Melia et al., 2016; Barber et al., 2018; Smith and Stephenson, 2013; Mudryk et al., 2021; Gascard et al., 2017). Moreover, forecasts of other variables also benefit from a well-represented sea ice component. For instance, Day et al. (2014) show that more accurate sea ice forecasts through sea ice thickness initialization can improve seasonal 2 m air temperature predictions. For these reasons, skillful and reliable sea ice forecasts, including for sea ice drift, are gaining importance.

Initialized predictions inevitably come with errors and uncertainty. Errors arise from physical models being simplified representations of reality, incomplete knowledge of the initial conditions, and inevitable chaotic error growth (Lorenz, 1969, 1975), which gave rise to ensemble forecasting. Here, we therefore differentiate between *errors*, which one should strive to reduce, and *uncertainty*, which must be represented (and communicated) adequately. Both of them act on the forecast accuracy, or skill.

An initialized forecast is commonly considered skillful as long as its accuracy is higher than some chosen benchmark, for example a climatological reference forecast. The lead time at which the initialized forecast ceases to be significantly more accurate than the – usually less sophisticated, and thus computationally cheaper – benchmark forecast, is the forecast skill horizon. This definition is not well suited when it comes to comparing skill estimates between different studies because statistical significance is a function of sample size; a study that utilizes a very large sample may detect significant skill even where the signal-to-noise ratio is small, and the practical relevance may be questionable, whereas a study that utilizes a small sample may fail to detect skill at practically still relevant levels. The same reasoning holds for estimates of potential predictability and, correspondingly, the predictability horizon. This ambiguity needs to be kept in mind in the following overview.

Recent studies have assessed both the current skill of sea ice drift forecasts and the potential to improve them, mostly for the Arctic Ocean. Grumbine (2013) finds ice drift trajectories obtained from a linear drift law within an operational ice drift model (Grumbine, 1998) to be skillful up to 16 d. Rabatel et al. (2018) show for neXtSIM (Rampal et al., 2016) that the representation of internal stress reduces forecast errors compared to a free drift model. Schweiger and Zhang (2015) investigate errors of short-term drift forecasts from the coupled-ice ocean model (MIZMAZ, forced with atmospheric forecasts) and find speed and position forecasts to remain skillful throughout the chosen integration time of 9 d. They suggest wind velocity forecast improvements to be the most promising approach to improve ice drift forecasts. Hebert et al. (2015) analyze short term (1-7 d) forecasts in an operational model (ACNFS, Posey et al., 2010) and find a seasonal bias of forecast errors in the Arctic; modeled drift speeds are slower than speeds obtained from buoy trajectories in summer, and faster in winter. For the marginal ice zone in the Southern Ocean, de Vos et al. (2021) find seasonal differences for short-term forecast accuracy (skill lower in winter than in spring) using buoy observations. Palerme and Müller (2021) show that the mean absolute error of operational 10 d ice drift forecasts from TOPAZ4 can be reduced using newly developed calibration methods based on supervised machine learning, and Andersson et al. (2021) present a probabilistic deep learning forecasting system producing more accurate seasonal forecasts of the summer ice state than SEAS5 (Johnson et al., 2019), especially for extreme sea ice events.



In these forecast skill assessments, all aforementioned sources of errors and uncertainty are imprinted on the forecast accuracy. While the forecast skill horizon can be pushed towards longer lead times by more sophisticated data assimilation and

forecast calibration methods, more efficient and more accurate numerical integration schemes and improved model physics, the uncertainty due to the sensitive dependence on the initial conditions is an inherent feature of the climate system and introduces an upper limit for forecasting skill (see, for instance, Collins, 2002; Hoskins, 2013). It is possible to estimate this inherent uncertainty in so-called perfect-model experiments, in which a model is not used for predicting reality, but predicting itself. Emulating perfect knowledge of the initial state or a complete and perfectly assimilated observational network, the initial

conditions are generated by the respective model in this approach. Then the divergence of the ensemble regarding a selected climate variable (predictand), with the members only differing by very small perturbations of the initial conditions, serves as an estimate for the growing uncertainty of the prediction due to the inherent chaos in the system. Analogous to the forecast skill horizon, here we call a variable (potentially) predictable up to a certain lead time as long as the uncertainty of an initialized ensemble forecast due to chaotic error growth is smaller than the expected climatological uncertainty, that is, the uncertainty

of an ensemble forecast constructed from independent years simulated by the same model with constant mean climate and variance.

While we are not aware of studies about the potential predictability of sea-ice drift, the predictability of various other sea ice properties has been assessed in a number of recent studies. The following studies are all based on data from the APPOSITE project (see Sect. 2) and ancillary simulations if not stated otherwise, as is the present study.

Sea ice extent and sea ice volume were found to be predictable up to three years in four coupled general circulation models (GCMs) by Tietsche et al. (2014), with sea ice volume being more predictable than extent. Sea ice volume predictability was found to depend on the initial month: predictive skill for sea ice volume declines much faster for ensemble forecasts initialized in May, opposed to those initialized in January or July (Day et al., 2014).

Cruz-García et al. (2019) complement these studies by a regional analysis of sea ice area, extent, and volume predictability.

For instance, while sea ice area in the peripheral seas exhibits significant predictability in winter, predictability is low in the other seasons, caused by the low predictability of the dynamic (two-dimensional) location of the sea ice edge, which was found to be predictable at least up to six months (Goessling et al., 2016). Goessling and Jung (2018) introduce a probabilistic verification metric for the location of the sea ice edge (the spatial predictability score, SPS) and report sea ice edge predictability up to ten months for the AWI-CM1 model. Zampieri et al. (2018) apply the SPS to a set of operational models for assessing

actual skill of sea ice edge forecasts and find a wide range of skill across the models, ranging from no skill even at initial time to skillful forecasts up to 1.5 months, which also highlights the gap between potential predictability and current forecasting skill.

Note that these studies use different metrics for predictability, tailored to the predictand in question. Therefore, the term "predictable up to" is not only influenced by sample size, as mentioned above, but also implicitly includes choices on the

metric and climatological normalization (see Sect. 2) and always refers to the predictability within a specific model. Also note that predictability from perfect model experiments is merely an estimate for the upper limit of forecast skill, as the real climate system could be more predictable than perfect model experiments suggest (Eade et al., 2014).





The substantial derivative, that is, the material change of sea ice momentum with time, can be described by

$$\frac{\mathrm{D}(\rho h \boldsymbol{u})}{\mathrm{D}t} = -\rho h f \boldsymbol{k} \times \boldsymbol{u} - \rho h g \nabla \psi + \boldsymbol{\tau}_a + \boldsymbol{\tau}_o + \nabla \cdot \boldsymbol{\sigma}, \tag{1}$$

where $\rho h$ is the inertial mass, $-f\boldsymbol{k} \times \boldsymbol{u}$ is the Coriolis acceleration with unit vector $\boldsymbol{k}$ in vertical direction, $-g\nabla\psi$ is an acceleration term from sea surface tilt with gravitational constant $g$, $\boldsymbol{\tau}_a$ and $\boldsymbol{\tau}_o$ are surface drag from wind and ocean currents, and $\boldsymbol{\sigma}$ is the stress tensor, which describes how the ice responds to forces trying to deform it (internal stress). A scale analysis (see, for instance, Leppäranta, 2011) reveals that, on time scales longer than a few hours, internal stress and atmospheric and oceanic drag are the terms of leading order.

While the wind is the primary driving force setting the ice into motion, the ocean drag mainly acts to counteract the wind-induced motion. This simplified relation does not hold where notable (sub-)surface ocean currents occur. Nevertheless, particularly in the open ocean and on the time scales of days, more than two thirds of the ice velocity variance is explained by geostrophic winds (Thorndike and Colony, 1982). The relation is weaker towards coastal areas, where other processes gain relevance, such as ice stress. Recent studies on potential explanations for an observed acceleration of Arctic ice speed in the last decades emphasize the importance of ice stress and strength for sea ice kinematics, particularly the effects of thinning ice and cracks not refreezing in a warming Arctic (Olason and Notz, 2014; Spreen et al., 2011; Rampal et al., 2009; Zhang et al., 2012).

It thus appears plausible that the predictability of near-surface wind determines the predictability of ice drift to a large part. However, also ocean drag, which is influenced by ocean currents, and the rheological response, which is influenced by ice thickness and concentration, affect the persistence and variability of sea ice drift and thus its predictability. Given that ocean currents as well as ice thickness vary on longer timescales than atmospheric winds, we hypothesize that the motion of sea ice is more predictable than the (near-surface) wind field. Moreover, as atmospheric circulation patterns, ice thickness, and ocean currents vary regionally and seasonally, we expect to find regional and seasonal differences in ice drift predictability. The main goal of this work is testing these hypotheses and quantifying the relative importance of the described effects.

With this study, we provide the first estimates of initial-value predictability of Arctic sea ice drift in global climate models. We use a suite of perfect-model ensembles following a common experimental protocol, which further enables an assessment of the model diversity regarding inherent ice drift uncertainty. We present the topic in both a Lagrangian perspective, i.e. the predictability of a time-dependent position of a virtual buoy, and an Eulerian point of view, i.e. the predictability of ice drift vectors at a fixed position, and explore possible explanations for regional and seasonal differences, while considering lead times from days to a few months. The Eulerian perspective allows us to compare ice drift predictability directly with corresponding wind predictability, and thus to quantify if and to what degree ice drift is more predictable than the wind. Our results will help to clarify how different processes act on sea ice drift predictability and put current forecast skill into perspective.

This manuscript is structured as follows. In Sect. 2 we introduce the so-called perfect-model approach, together with the data set and methods in use. We assess the individual model background climates in Sect. 3. The predictability of (Lagrangian) target positions and (Eulerian) velocity vectors is presented in Sect. 4.1 and then interpreted in the light of near-surface wind





speed (Sect. 4.2) and initial ice thickness (Sect. 4.3). We then discuss our results critically before summarizing and drawing conclusions.

## 2 Data and methods

### 2.1 APPOSITE data set

As part of the *Arctic Predictability and Prediction on Seasonal-to-Interannual TimEscales* (APPOSITE) project, perfect-model experiments have been carried out with seven general circulation models (Day et al., 2016). The project has been designed for an assessment of model diversity in Arctic predictability on seasonal to interannual time scales. Due to the common experimental setup, all variation across the models can be attributed to the different representation of physical processes, discretizations and integration schemes, which allows a direct model comparison. In this study, we analyze four of the seven participating models:

– AWI-CM1 (Sidorenko et al., 2015), a climate model run and developed by the Alfred Wegener Institute for Polar and Marine Research (Germany),

   – GFDL-CM3 (Griffies et al., 2011; Donner et al., 2011), from the Geophysical Fluid Dynamics Laboratory (USA),

   – HadGEM1.2 (Johns et al., 2006; Shaffrey et al., 2009), the Hadley Centre Global Environment Model from the Met Office (UK), and

– MPI-ESM (Notz et al., 2013; Jungclaus et al., 2013), an earth system model developed by the Max Planck Institute for Meteorology (Germany).

   Simulations from MPI-ESM and GFDL-CM3 were submitted to the Coupled Model Intercomparison Project Phase 5 (CMIP5) in the same configuration as for the APPOSITE project. Note that AWI-CM1 and MPI-ESM share the same atmospheric component, ECHAM6. In some other publications, AWI-CM1 may be referred to as E6F, a temporary former 145 name.

   For each of the selected models, a control simulation of at least $N_{\mathrm{ctrl}} = 200$ years integration time is available, with at least 100 years for spin-up beforehand. As the predictability of the climate system is sensitive to the mean climate (DelSole et al., 2014) and because secular trends make it more difficult to quantify predictability, all used simulations were performed using a fixed radiative forcing representative of the recent past. These control simulations provide statistical properties of the 150 individual model climates. Depending on the model, up to $N_{\mathrm{init}} = 18$ ensemble forecasts with up to $N_{\mathrm{mem}} = 16$ members were initialized, using initial conditions from the control run (see Table 1). The members were generated by adding different realizations of uncorrelated white noise with a standard deviation of $10^{-4}$ K to the sea surface temperature. The divergence of the model members is directly attributable to the amplification of this virtually infinitesimal perturbation. The common protocol required initializations on 1st July; three out of four models provided additional initializations on 1st January. This enables an 155 analysis of the seasonal dependence of the growth of uncertainty. To sample the background variability of the respective model climate, the initialization years were distributed over the whole control run.



**Table 1.** Brief summary of the APPOSITE simulations used in this study. The columns are, from left to right, model name, number of control simulation years ($N_{\mathrm{ctrl}}$), number of ensemble members ($N_{\mathrm{mem}}$), number of initialized predictions ($N_{\mathrm{init}}$) and the initialization months.

| Model | $N_{\mathrm{ctrl}}$ | $N_{\mathrm{mem}}$ | $N_{\mathrm{init}}$ | init. months |
|---|---|---|---|---|
| AWI-CM1 | 200 | 9 | 18 | Jan, Jul |
| GFDL-CM3 | 200 | 16 | 8 | Jan, Jul |
| HadGEM1.2 | 249 | 16 | 10 | Jan, Jul |
| MPI-ESM | 200 | 9 | 12 | Jul |

We use daily averaged model output for sea ice velocity, sea ice concentration (SIC) and sea ice thickness (SIT) from all four models, and near-surface (10 m) wind velocities additionally provided for AWI-CM1 and HadGEM1.2. Unfortunately, daily ocean velocities are not available, so that we need to restrict our analyses to the aforementioned variables.

## 2.2 Trajectory calculation and selection

We calculate kinematic trajectories for virtual buoys (targets) from the velocity fields with a numeric integration method of second-order accuracy. Trajectories $\boldsymbol{x}(t)$ are solutions of the ordinary differential equation

$$\dot{\boldsymbol{x}}(t) = \boldsymbol{u}(\boldsymbol{x}, t) \tag{2}$$

with an initial condition $\boldsymbol{x}_0 = \boldsymbol{x}(t_0)$, where $\boldsymbol{u}(\boldsymbol{x}, t)$ is the ice velocity field and $t$ the time.

Assuming that the given initial value problem fulfills the requirements for the Picard-Lindelöf theorem, unique analytical solutions are given by

$$\boldsymbol{x}(t) = \boldsymbol{x}_0 + \int\limits_{t_0}^{t} \boldsymbol{u}(\boldsymbol{x}(s), s)\, \mathrm{d}s, \tag{3}$$

which aptly illustrates how the two predictands we focus on, Eulerian velocity $\boldsymbol{u}$ and Lagrangian target position $\boldsymbol{x}$, are connected.

As AWI-CM1 computes on an unstructured triangular grid, we implemented the integration method for that type of mesh and applied it to curvilinear and rectilinear grids by cutting the quadrilateral boxes into two adjacent triangles. The method uses an adaptive time step which ensures that no triangular elements are skipped during one time step, thus avoiding a loss of velocity information. We use barycentric coordinates for the spatial interpolation between nodes and a common linear interpolation in time. The trajectory tool is a side product of this publication and made openly available in the R-package *spheRlab* (https://github.com/FESOM/spheRlab). Before the trajectory computation, we rotated the coordinate system such that the North Pole lies on the equator to avoid a singularity in the spatial domain. We interpolate the output onto a 0.5 d time grid for facilitating the subsequent analyses, which is also the default for the adaptive time step.



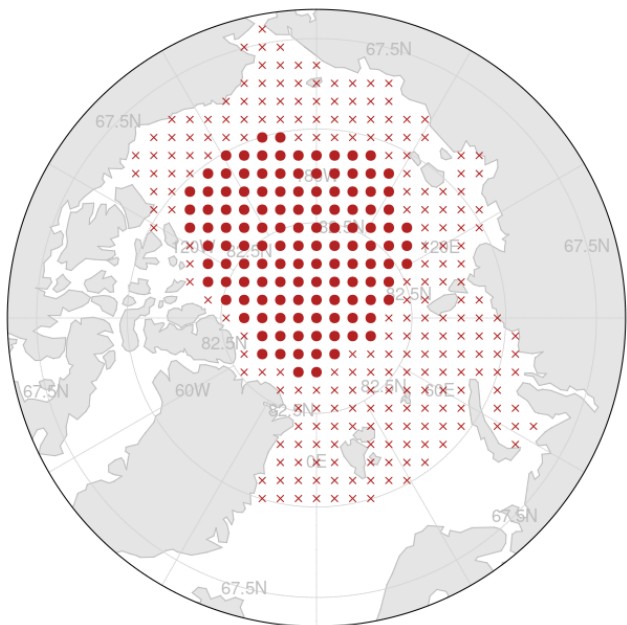

**Figure 1.** Map of all initial positions for trajectory computation (red crosses and circles). The initial positions with a solid red circle constitute the common subset of initial positions for all models after the filtering process outlined in Sect. 2.2.

We use 347 initial positions distributed evenly over the Arctic Ocean (see Fig. 1), which allows us to assess regional differences of predictability. The distance between two neighboring initial positions is approximately 160 km. Trajectories were computed for all aforementioned models, for all available control run years and all initialized ensemble predictions, always initializing the trajectories on 1st January (all models) and 1st July (all models except MPI-ESM). The maximum integration time is 120 d and the computation of a trajectory is stopped prematurely when the virtual buoy enters a region with sea ice concentration lower than 0.15 or exits the ocean model domain. Therefore, a fraction of targets gets lost over time, particularly from those initialized close to the ice edge and Fram Strait. To account for that, we select initial positions for the analysis based on the following filtering criteria:

1. *Residence time in ice cover*: We exclude all trajectories not covering at least 90 d (45 d) for January (July) initializations.

2. *Complete ensembles*: If an ensemble prediction for a given target is missing any member after the previous filtering step, we remove the given ensemble from the analysis.

3. *Minimum number of initializations*: Dependent on the respective model, there are between eight and eighteen initializations (i.e., ensembles) per initial target position. We remove all initial positions from the analysis which do not offer at least eight complete ensembles. This holds for both the initialized ensemble predictions and the climatological reference ensembles (see Sect. 2.3).





The longer the selected time period for analysis (in the first step), the more trajectories get removed, and the smaller the
spatial coverage becomes. The chosen thresholds are therefore a compromise within the trade-off between spatial and temporal
coverage. For the comparison of time series of spatially integrated quantities between the individual models we use the common
subset of targets after the outlined filtering process, i.e. the intersection of all available targets from the different models of the
initialized prediction and control runs (see also Fig. 1), that is, 126 targets in total. As a consequence, the marginal ice zone
is not represented in these parts of the analyses, and the inter-model seasonal ice cover dictates the shape and location of the
common subset.

## 2.3   Measures of predictability

In this study, we determine and analyze the potential predictability of Lagrangian target positions and Eulerian velocity vectors,
which are both two-dimensional quantities. We express the predictability as the ratio of the uncertainty of the ensemble mean
of the initialized prediction to a reference ensemble constructed by drawing randomly from the climatological background (see
below). It is therefore necessary to measure the uncertainty of an ensemble prediction. For scalar quantities, such as temperature
or ice speed, the normalized root mean squared error (NRMSE, see Collins (2002)) is used in various predictability studies. To
account for the bivariate character of position and velocity vectors, we chose a different approach here, which we exemplify in
the following for velocity vectors.

For a given ensemble of velocity vectors at a given position and lead time, we determine the variance ellipse. Our measure
for the uncertainty is then the length of the semi-major axis, which is the spectral norm of the covariance matrix of the velocity
vectors. This also enables an analysis of the axis ratio and thus the anisotropy of the uncertainty. The uncertainty of initialized
forecasts is then given by the mean of all available initializations (at least eight, due to the filtering).

For the normalization, we construct a climatological reference ensemble by drawing randomly from the control run. For a
specific initial position and season from AWI-CM1, for example, we compile an ensemble of the same size as the initialized
predictions (i.e. $N_{\mathrm{mem}} = 9$ for AWI-CM1) by drawing $N_{\mathrm{mem}}$ random years from the control simulation, for the same season,
and calculate the uncertainty for this ensemble as before. We repeat this procedure 200 times and estimate the climatological
uncertainty by the mean uncertainty of those 200 randomly generated ensembles. This is performed for each initial position for
both January and July initializations.

The normalized uncertainty $\hat{\sigma}_{i,n}$ of the $i$th initialized ensemble for the $n$th target position is then given by

$$\hat{\sigma}_{i,n} = \frac{a_{i,n}}{\langle a_n^{\mathrm{clim}} \rangle}, \tag{4}$$

where $a_{i,n}$ is the semi-major axis length of the $i$th initialized ensemble, and $\langle a_n^{\mathrm{clim}} \rangle$ is the expectation value from the climato-
logical randomly generated ensembles (estimated by the mean); note that the explicit time dependence has been dropped for
simplicity.

We obtain a time series of normalized uncertainty for the whole common subset by averaging over the index $n$, which allows
using the different initializations as sample for testing hypotheses. For presenting geographical features of the uncertainty, we





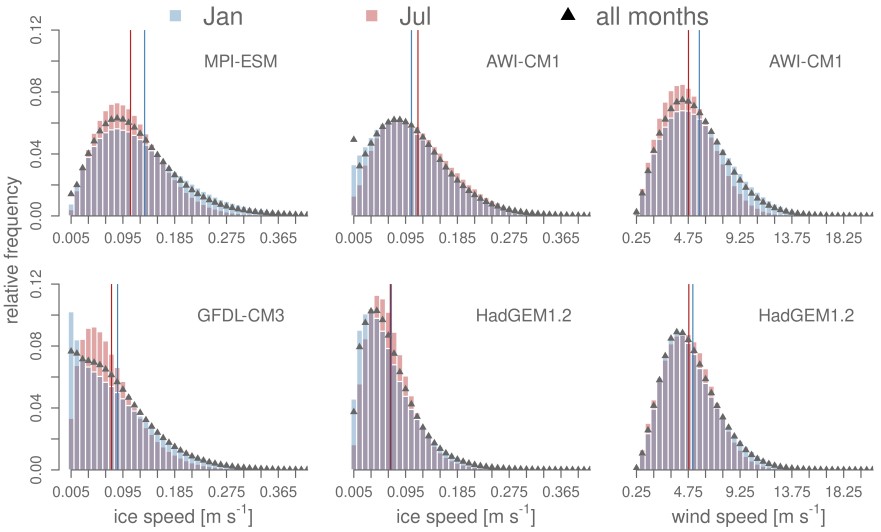

**Figure 2.** Relative frequency distribution for ice drift speed (left two plots in each row) and near-surface wind speed (right panels) on the common subset and only including days with SIC $\geq 0.15$ and SIT $\geq 1$ mm, based on the respective control simulation. Blue bars show data for all January months in the control run, red bars for July, black triangles include the data from all months. The blue (red) vertical lines show the mean speed for January (July). The bin size is $0.01$ m s$^{-1}$ for ice speed and $0.50$ m s$^{-1}$ for wind speed.

average over $i$, obtaining $\hat{\sigma}_n$. A value of $\hat{\sigma}_n = 1$ indicates that the uncertainty of the initialized prediction for a given position has reached the level of the climatological uncertainty, which implies a complete loss of information from the initial conditions.

Note that potential predictability is often defined via $1 - \hat{\sigma}_n$, we however prefer the notion of an increasing uncertainty of the initialized forecast relative to the climatological spread and will use $\hat{\sigma}_n$ throughout this work.

We follow the same approach for the position vectors, only that we first use an orthographic azimuthal projection onto the
plane tangential to the barycenter of the point cloud given by a single ensemble prediction from a given initial position, for obtaining two-dimensional Cartesian coordinates (in km).

## 3 Model climate and ice drift variability

Here we present main characteristics of climatological variables related to sea ice momentum and motion, namely sea ice thickness, ice velocity and speed, and, where available, wind. We further present key aspects of the interannual ice drift
variability, based on the calculated trajectories. We focus on the results of January and July, as the ensemble predictions were initialized in these months.

An assessment of average Arctic sea ice concentration, sea ice extent and sea ice volume of the APPOSITE simulations is provided by Day et al. (2016), where also maps of average ice thickness for the months March and September can be found. Mean state and variability of the control simulations regarding those variables vary substantially between the different





models, with HadGEM1.2 exhibiting the largest mean sea ice extent and volume, followed by GFDL-CM3 and then, both at a
comparable level, MPI-ESM and AWI-CM1.

This order also holds for ice thickness (see Table A1). All models simulate the thickest ice towards the coast of the Canadian
Arctic Archipelago (CAA), although the spatial gradients are comparably small in MPI-ESM (Fig. A1). Tietsche et al. (2014)
find the SIT variability for GFDL-CM3, HadGEM1.2 and MPI-ESM in the APPOSITE simulations to be larger in coastal
areas, likely caused by surface wind variability and thus advective processes rather than thermodynamics.

For all models, we determined the ice speed distribution on the previously introduced common subset of initial positions
from the control run for the initialization months January and July (Fig. 2, left two columns). While ice speeds are larger in
January compared to July for GFDL-CM3, HadGEM1.2 and MPI-ESM, the mean ice speed in July is higher than in January
for AWI-CM1. Overall, annual mean ice speed on the common subset is highest for MPI-ESM ($0.12 \, \mathrm{m \, s^{-1}}$); this model is also
the only one where the frequency density at very low speeds drops almost to zero, even in winter, suggesting a smaller role
of ice rheology. The other models exhibit lower average drift speeds of $0.10 \, \mathrm{m \, s^{-1}}$ (AWI-CM1), $0.08 \, \mathrm{m \, s^{-1}}$ (GFDL-CM3)
and $0.07 \, \mathrm{m \, s^{-1}}$ (HadGEM1.2). For all models, the distribution is more positively skewed, with higher relative frequencies
of the lowest ice speeds in winter compared to summer, presumably caused by a stronger ice pack, albeit this effect is very
small in MPI-ESM. At the same time, MPI-ESM, GFDL-CM3 and HadGEM1.2 also exhibit an increase of the higher drift
speeds in winter, likely due to increased wind speeds. Higher wind speeds in winter compared to summer indeed occur in
both HadGEM1.2 and AWI-CM1, the two models for which daily wind data is available (Fig. 2, right column); it appears
likely that the same holds for the other two models. It is noteworthy that the higher winterly wind speeds in AWI-CM1 do not
translate into more frequent occurrence of high ice drift speeds in AWI-CM1 in winter. We speculate that this might be either
due to seasonal changes in wind direction that counteract the wind speed effect, e.g. because winterly winds might tend to blow
more often towards coasts, or due to a stronger influence of seasonal ice thickness variations on ice drift even in cases where
rheological effects do not much attenuate the ice drift. However, resolving these intricacies is beyond the scope of this work.

For the two models providing wind data, we investigate the climatological annual cycle of the relation of ice speed and
wind speed on the analysis grid. For AWI-CM1, there is a pronounced phase shift of monthly mean ice speed and wind speed:
while ice speed is lowest in April and maximum in August, wind speed is lowest in June and July and peaks in the cold
season between August and February (Fig. 3). We suggest that the phase relation of drift speed and wind speed in AWI-CM1
is determined by ice mobility. While wind speeds are maximal in winter, ice speed slows down with increasing thickness. With
the onset of melting season in spring, ice becomes weaker and more susceptible to wind stress. Albeit the wind speeds are
lowest in June and July, ice accelerates due to weakening of the ice pack. This phase relation may be modulated further by
seasonal changes in large-scale circulation patterns. For HadGEM1.2 ice speeds are significantly lower than in AWI-CM1 and
the amplitude of the seasonal wind speed cycle is much smaller.

The differences between all four models regarding the climatological ice drift speed, as well as the differences between
AWI-CM1 and HadGEM1.2 regarding the magnitude of the climatological wind forcing, already hint at potential differences
in the growth of uncertainty of ice drift predictions between models, seasons, and regions.





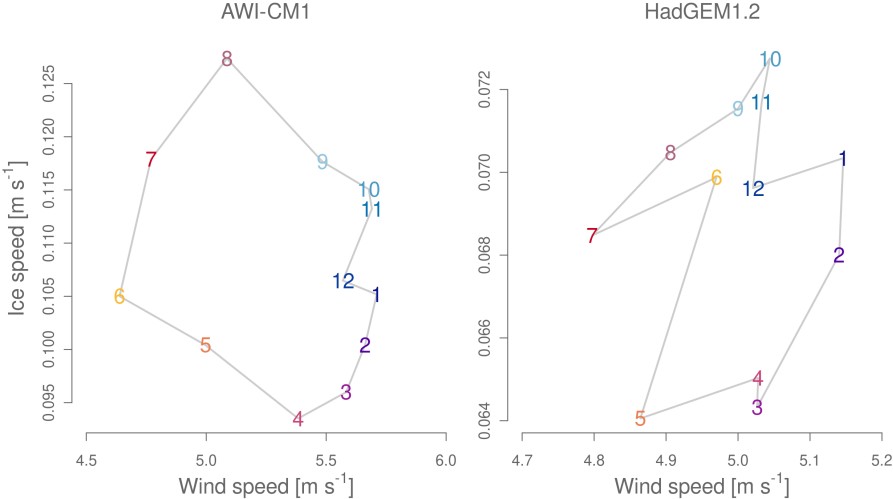

**Figure 3.** Annual cycles of monthly mean wind and drift speeds on the common subset of initial positions. The numbers indicate the respective months. Please note the strongly different scaling on the axes.

In the following, we consider the trajectories from the control simulations. For each initial position, we derive the orientation
and axis ratio of the variance ellipse for the climatological distribution of $N_{\text{ctrl}}$ target positions at 45 d lead time (Figs. 4 and 5), both for January and July. The anisotropy given by the axis ratio shows whether there is a preferred direction of uncertainty growth for an initial position. The models share some common features in that matter; ellipses are more eccentric towards the Northern and Eastern coast of Greenland and the coast of Alaska, and more circular (i.e. isotropic) in the open ocean. In July, eccentricity in the coastal areas decreases, likely a sign of increased mobility due to a weaker ice pack and/or more variability
in the wind direction. Together with the seasonal differences in the ice speed distribution, this is another factor that might lead to seasonality in ice drift predictability. Also note that the direction of maximum variance can vary from being parallel (e.g. in Fram Strait) to perpendicular (e.g. north of Greenland and CAA) to the mean motion.

The temporal evolution of the spatially averaged climatological reference uncertainty used for normalization (obtained by bootstrapping, see Sect. 2) for the Lagrangian target position follows approximately the shape of a square root function for the
examined lead time, similar to a Brownian "random walk" motion (Fig. A2). As already recognizable from the depicted ellipse size, the climatological uncertainty is highest for MPI-ESM. For HadGEM1.2 and GFDL-CM3, the climatological reference uncertainty is larger in January than in July, contrary to AWI-CM1. The July uncertainty of MPI-ESM follows closely the one from AWI-CM1.

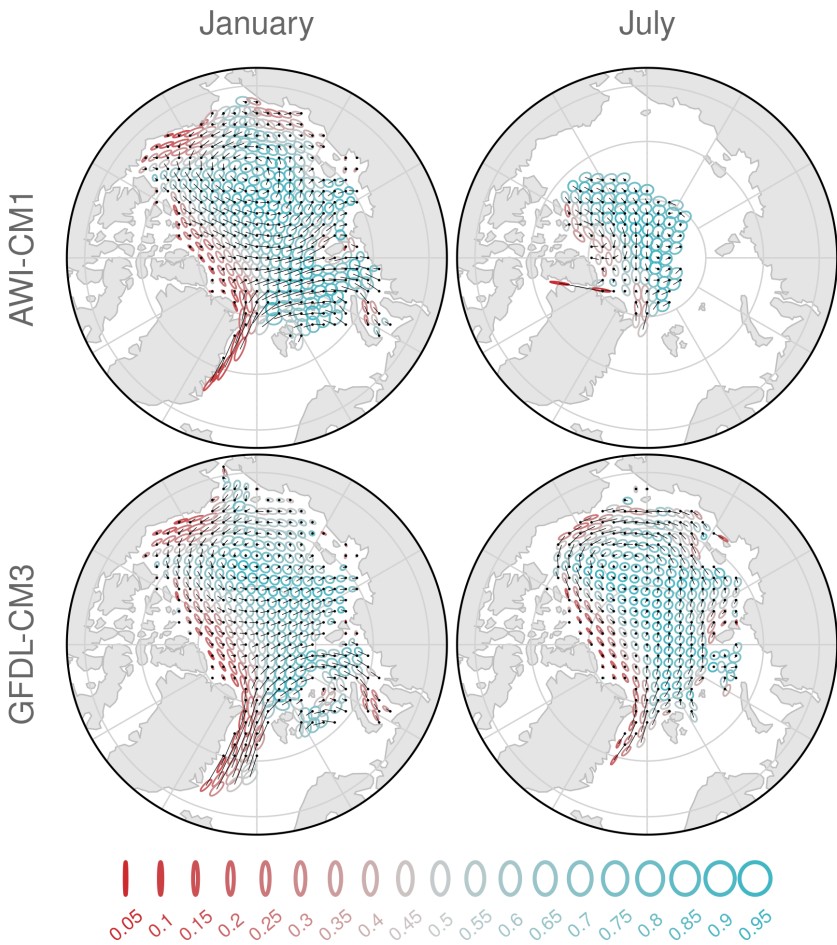

**Figure 4.** Variance ellipses (here scaled for showing 20 % confidence level, scale chosen for best visibility) for the target position at $t = 45$ d lead time, generated from trajectories from all available control run years; we only show those initial positions for which the targets of at least 180 single years remained in the ice cover up to $t = 45$ d. Each row shows ellipses for initializations in January (left) and July (right). The coloring denotes the axis ratio (anisotropy) of the uncertainty, the black dots show the corresponding initial positions, connected by thin black lines to the ellipse centers, which indicate the average positions after 45 d.

## 4 Predictability of sea ice drift

### 4.1 Ice drift predictability

#### 4.1.1 Lagrangian perspective

We begin with the analysis of the target position predictability, measured by the normalized spread (uncertainty) of the point clouds that correspond to individual lead times of the trajectory ensembles. In the following, we use the term uncertainty





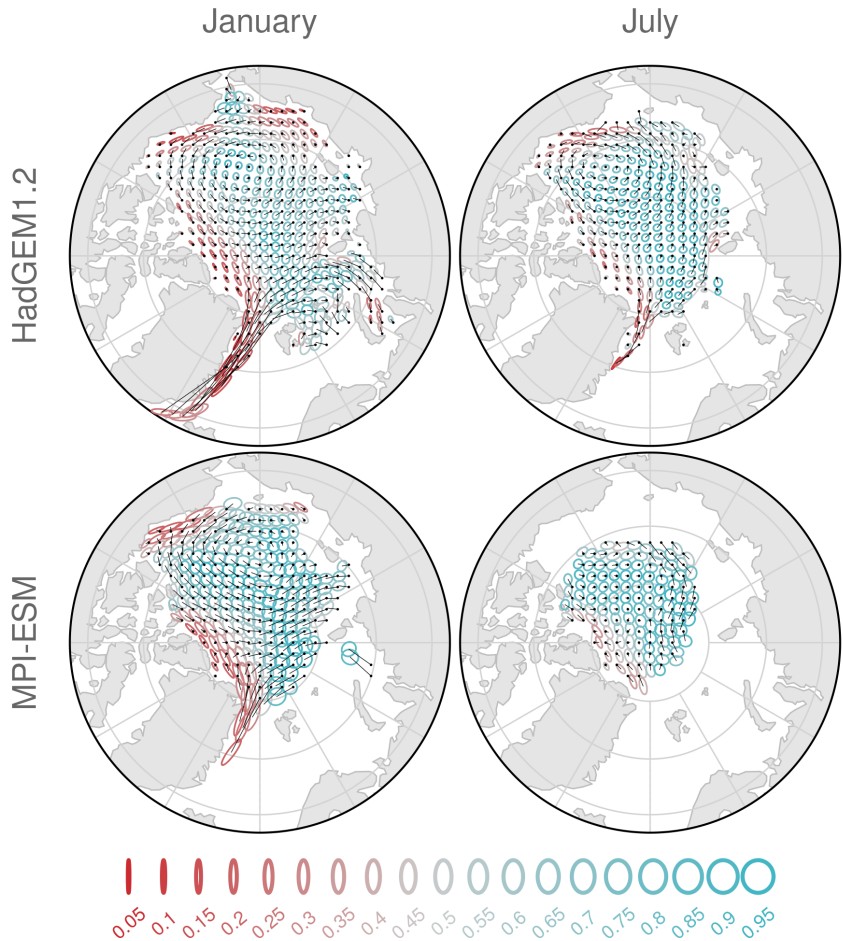

**Figure 5.** As Fig. 4, but for the models HadGEM1.2 and MPI-ESM.

interchangeably with normalized uncertainty. As explained above, a (normalized) uncertainty of 0 implies perfect predictability,

whereas a (normalized) uncertainty of 1 implies the complete loss of predictability.

The spatial characteristics are presented for 45 d lead time, which is the range of the summer trajectories. However, the patterns are changing only slowly over time and are thus representative for a broad time period from about 30 to (at least) 45 d lead time.

The uncertainty exhibits smooth spatial gradients in all models (Fig. 6). The spatial patterns differ between the models and

between summer and winter. In AWI-CM1 the uncertainty from January initializations grows more slowly along the northern coast of the CAA, the Beaufort Sea, the Chukchi Sea and the East Siberian Sea than in the open ocean, the Kara Sea, the Laptev Sea and the Barents Sea. While the uncertainty in the Beaufort Sea in GFDL-CM3 is also comparably low (with higher values





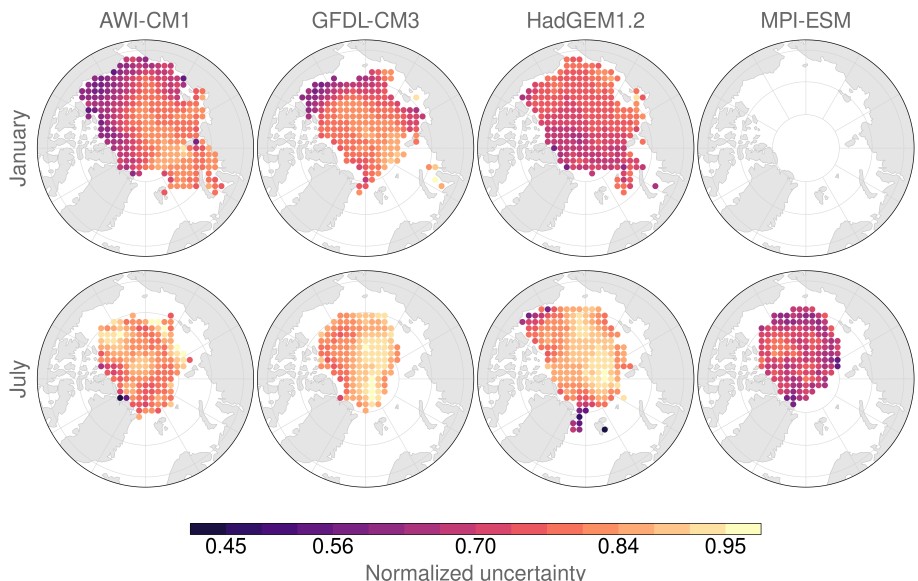

**Figure 6.** Normalized uncertainty of target position predictions for all initial positions that fulfil the selection criteria I-III (see <section>) at 45 d lead time. Top row: initializations on 1st January. Bottom row: initializations on 1st July. Note that there are no January initializations for MPI-ESM available.

in the open ocean), this model has a less pronounced North-South gradient towards Greenland and the CAA. HadGEM1.2 shows generally weaker spatial gradients in winter than AWI-CM1 and GFDL-CM3.

There are distinct regions with significantly higher uncertainty for July initializations than for January initializations in AWI-CM1, GFDL-CM3 and HadGEM1.2 (also Fig. 6), based on a one-sided unpaired t-test using the different start years as sample (not shown). For AWI-CM1, significant seasonal differences occur mainly north of the CAA, where the normalized uncertainty is low in winter. For the same area, there is no significant seasonal difference in GFDL-CM3, where seasonal differences occur mainly between 90 and 200° E. The majority of initial positions with January and July initializations in

HadGEM1.2 are significantly more uncertain in July, except for the Beaufort Sea and the outflow region towards Fram Strait. Spatial gradients in HadGEM1.2 are larger in summer. Interestingly, the magnitudes of normalized uncertainty for MPI-ESM in summer matches the range of the other models in winter, which follows from the fact that the normalized uncertainty in MPI-ESM is generally lower than in the other models in summer.

     Constraining the examined region to the common analysis grid, we find that the spatially averaged normalized uncertainty

has qualitatively the same temporal evolution in all four models: a slow increase of uncertainty in the first days, a relatively steep increase within roughly four weeks lead time, and then a deceleration of normalized uncertainty growth (Fig. 7). This deceleration can be understood as a phase during which the uncertainty of the initialized predictions grows at a similar rate like the climatological reference.





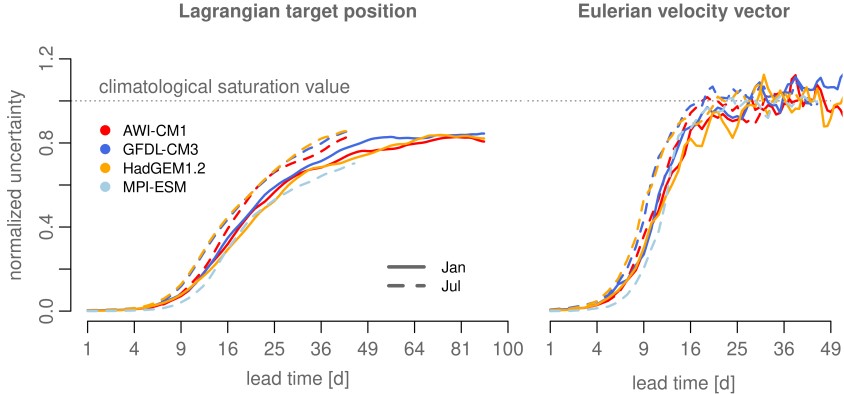

**Figure 7.** *Left*: Temporal evolution of normalized uncertainty of Lagrangian target position for all models, spatially averaged on the common subset of initial positions. Solid lines show the January initializations, dashed lines July. The horizontal dashed line indicates the climatological saturation value. Please note the square root horizontal axis. *Right*: Same as left, but for the Eulerian ice velocity.

Despite the qualitatively different spatial distributions, the averaged normalized uncertainty is very similar across AWI-
CM1, GFDL-CM3 and HadGEM1.2 (see Table 2 for numerical values). The uncertainty in MPI-ESM for July initializations
is relatively low, and resembles more the winter uncertainty of the other models than their respective July initializations. This
is despite the facts that the climatological reference uncertainty for July initializations is similar between AWI-CM1 and MPI-
ESM (Fig. A2), that both models share the same atmospheric component, and that they have quite similar seasonal cycles of sea
ice extent and volume (Day et al., 2016) and, to a slightly lesser extent, sea ice thickness. This suggests that the significantly
different behavior of AWI-CM1 and MPI-ESM regarding the growth of the (non-normalized) uncertainty of the initialized
ensembles should be either due to differences in the ocean- and sea ice components of the models, or related to a delayed
divergence of the atmospheric states in MPI-ESM (see Sect. 5).

None of the models reaches the climatological saturation value for target position uncertainty within 90 d (45 d) lead time in
January (July), and the growth appears to level off for long lead times for the winter initializations. This seemingly asymptotic
behavior indicates that the growth rates of initialized and randomly generated ensembles equalize at some point, at least for
the depicted time period. The spread difference between the initialized and climatological trajectory ensembles that develops
during the first weeks is maintained throughout the analyzed forecast range. One problem that hinders the analysis of longer
lead times is the inevitable loss of trajectories, so we cannot easily extend the examined time period without further reducing
the spatial analysis domain. Nevertheless, we follow three approaches for obtaining deeper insights: (i) an additional point of
view – the Eulerian perspective –, (ii) an additional experiment where we release targets at the original initial positions, but
at a later time when the model forecast fields have already advanced further ("lagged initialization"), and (iii) an additional
experiment with longer integration time.





**Table 2.** Mean normalized uncertainties on the common set of initial positions at 45 (12) days lead time for the Lagrangian (Eulerian) perspective with one standard error.

| Model | LAGR, Jan (45 d) | LAGR, Jul (45 d) | EULE, Jan (12 d) | EULE, Jul (12 d) |
|---|---|---|---|---|
| AWI-CM1 | $0.75 \pm 0.02$ | $0.84 \pm 0.02$ | $0.59 \pm 0.05$ | $0.65 \pm 0.05$ |
| GFDL-CM3 | $0.79 \pm 0.04$ | $0.86 \pm 0.02$ | $0.65 \pm 0.13$ | $0.78 \pm 0.05$ |
| HadGEM1.2 | $0.73 \pm 0.03$ | $0.87 \pm 0.04$ | $0.54 \pm 0.04$ | $0.80 \pm 0.06$ |
| MPI-ESM | - | $0.70 \pm 0.01$ | - | $0.52 \pm 0.03$ |

Regarding the last point, we calculated trajectories for AWI-CM1 with 240 d integration time, starting on 1st January, and only chose those initial positions which cover the entire time period, again using the established filtering criteria (except for
the longer time period). We find that the target position uncertainty eventually saturates within less than 190 days (Fig. A3). The first two points are the subject of the following sections.

### 4.1.2 Eulerian perspective

Instead of following virtual ice parcels on Lagrangian trajectories like in the previous section, we perform a similar analysis for sea ice drift velocity vectors at fixed positions here. This enables a direct comparison with the near-surface winds at the
same locations (see Sect. 4.2).

As for the Lagrangian point of view, all models exhibit spatial gradients of normalized uncertainty also for the drift velocity vectors (not shown). These gradients are smooth, though intermittent over time, in such that qualitative regional characteristics change within days. Significantly larger uncertainties for summer initializations can only be detected in HadGEM1.2, as comparably large interannual variability renders any differences of the mean uncertainties statistically insignificant for AWI-CM1
and GFDL-CM3.

In contrast to the Lagrangian perspective, where a considerable level of predictability appears to remain at comparably long lead times, from the Eulerian perspective the normalized uncertainty reaches the climatological uncertainty for all models in both seasons within less than three to four weeks (Fig. 7, right panel), and the uncertainty fluctuates around the saturation value thereafter. The largest difference between the models occurs between ten and sixteen days lead time (see again Table 2 for
selected numerical values).

### 4.1.3 Lagged-initialization experiment

As trajectories are obtained by applying a time integral to the velocity field (Eq. 3), the Lagrangian and Eulerian perspectives are naturally tightly connected (at least as long the targets remain in the vicinity of their initial positions). We highlight this connection with another experiment, in which we delay the initialization of the trajectories relative to the initialization of the
original ensemble simulation by 4, 7, 10, 15 and 20 days. As the climatological uncertainty has to be calculated for each



different lag time, and the bootstrapping method is computationally expensive, we estimate the climatological uncertainty only by 100 random draws for each initial position.

The initial normalized uncertainty of the lagged trajectories matches the uncertainty of the Eulerian velocity vectors at the respective original lead time very closely (Fig. 8), which can be explained as follows. Imagine we had used the explicit Euler

method for the trajectory calculation; the variance ellipse of target positions after the first time step would then be equivalent to the ellipse of the velocity vectors, merely scaled by time step size. However, the different scaling is compensated by the normalization, so that a close initial correspondence of the normalized uncertainty follows directly.

The uncertainty of the lagged Lagrangian trajectories then grows more slowly than for the Eulerian velocity vectors, bounded below from the uncertainty of the non-lagged trajectories. Figuratively speaking, the time integration leads to an "inheritance"

of the low uncertainty from early time steps for the Lagrangian trajectories, whereas subsequent time steps in the Eulerian perspective do not transport information in time. As displacements of buoys within a few time steps are small compared to the scale of spatial coherence of the ice velocity field in the analyzed models, the Eulerian and Lagrangian perspective tell a similar story – yet at a different pace. This feature may raise confidence that properties of the Eulerian ice velocity predictability can be partly transferred to the Lagrangian target position predictability and vice versa.

Finally, this experiment displays how an initial uncertainty of the velocity field is influencing the growth of uncertainty of the target position, which has practical consequences for operational forecasts; the smaller the initial uncertainty of the ice velocity, the later does the uncertainty growth rate of initialized forecasts attain the climatological growth. This underlines the important role of data assimilation.

In summary, we find that the spatial distribution of normalized target position varies between the models and, within the

same model, also between summer and winter initializations. For the Lagrangian perspective, normalized uncertainties do not saturate within the considered time periods, albeit an experiment with longer integration times shows that this happens at even longer lead times. The uncertainty of Eulerian velocity vectors on the other hand saturates within less than four weeks for all investigated models, while exhibiting intermittent spatial characteristics. A lagged initialization experiment highlights the close connection between both points of view. It furthermore illustrates how the target position uncertainty is inherited from the

Eulerian velocity vector uncertainty. In the following, we examine what drives the observed regional and seasonal differences.

### 4.2 Relation to near-surface wind

Wind forcing is known to be one main driver of ice drift variability, particularly in the open ocean, and it is therefore likely to play a key role in ice drift predictability. The APPOSITE data set includes daily average near-surface wind velocities for two of the considered models, AWI-CM1 and HadGEM1.2, of which we make use in this section. We calculate the normalized

uncertainty for the two-dimensional near-surface wind vectors in Eulerian perspective like for the ice velocities before.

Indeed, we find a close correspondence of the temporal evolution of wind and ice drift vector uncertainty (Fig. 9). However, there is a time period of more than two weeks when the ice uncertainty is significantly lower than wind uncertainty for AWI-CM1 in both January and July initializations (determined by a one-sided paired t-test, $95\%$ level). This is considerably less evident for HadGEM1.2. For this model the uncertainty of ice drift for January is generally lower than the one for near-surface



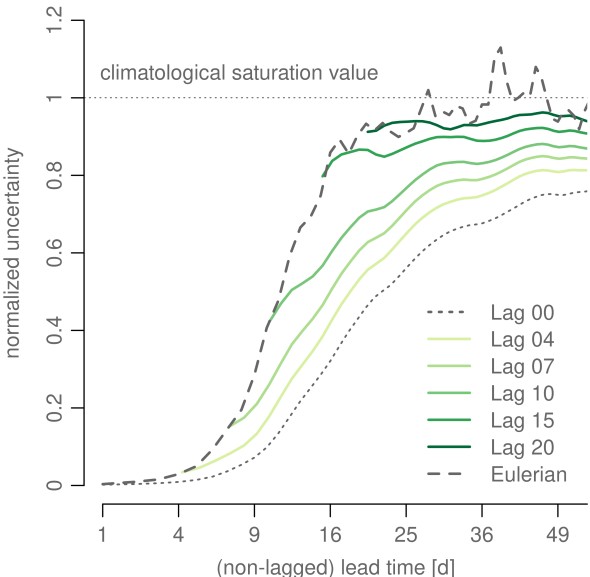

**Figure 8.** Lagged-initialization experiment with AWI-CM1: We set virtual buoys at the same initial positions as before, but delay the release by 4, 7, 10, 15 and 20 days. The colored lines show the normalized uncertainty on the non-lagged time axis. The dashed lines show the uncertainty for the Eulerian perspective and the dotted line the uncertainty of non-lagged Lagrangian trajectories ("Lag 00"). The initial uncertainty of the lagged Lagrangian trajectories matches closely the uncertainty in Eulerian perspective at the respective time of buoy release.

wind, yet the differences are only significant between nine and sixteen days lead time, considering only the time before first saturation. In July, the differences are only marginal. However, the less significant differences in HadGEM1.2 compared to AWI-CM1 do not necessarily reveal an actual model difference; the smaller number of initializations and thus the smaller sample size for HadGEM1.2 (10) compared to AWI-CM1 (18) could also explain the differences in statistical significance.

The spatial distribution of wind vector uncertainty, albeit being intermittent as well, matches closely the patterns of the
uncertainty of ice drift vectors (not shown). Moreover, there is a significant positive correlation of wind vector uncertainty and ice vector uncertainty at most initial positions in both models and both seasons, again taking the available number of initializations per initial position as sample (Fig. A4). That is, lower (higher) uncertainty of wind vectors coincides with lower (higher) uncertainty of ice drift vectors at a fixed position, and a considerable fraction of the variation of ice drift uncertainty can be attributed to the near-surface wind predictability. On the common subset of initial positions, the mean correlation
coefficient (average from Fisher z-transformed data, then back-transformed) for $t = 12$ d lead time is 0.79 (0.93) for January (July) for AWI-CM1, and 0.76 (0.92) for HadGEM1.2. After 28 d, the correlation coefficients are 0.79 (0.90) and 0.86 (0.87) for AWI-CM1 and HadGEM1.2, respectively. Still, there must be other processes leading to the slightly lower uncertainty of





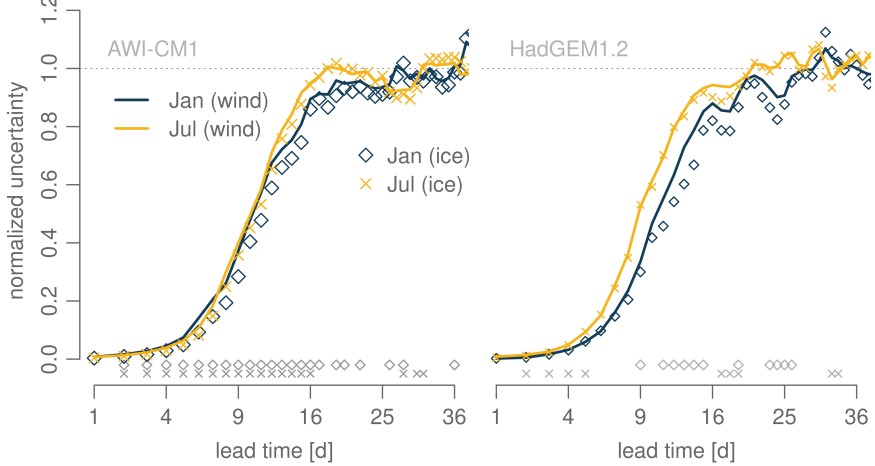

**Figure 9.** Normalized uncertainty of near-surface wind vectors in Eulerian perspective for AWI-CM1 (left) and HadGEM1.2 (right). Solid (dashed) lines denote the mean for January (July) initializations, and the normalized uncertainty of Eulerian ice drift vectors is added for comparison as diamonds (crosses) for January (July). The horizontal gray dotted line marks the climatological saturation value of 1. The diamonds and crosses above the horizontal axis denote those lead times for which the uncertainty for the ice velocity vectors is significantly lower than the uncertainty for wind vectors (one-sided paired t-test, 0.95 level).

the Eulerian ice drift, thus acting as a source of predictability against the sink imposed by the near-surface winds. One possible source of additional predictability is linked to the ice state, which is the subject of the following section.

## 4.3 Relation to initial ice state

Here, we examine the relation of initial ice thickness and the uncertainty of the target position (Lagrangian perspective). For each initial position, we calculate the correlation coefficient for initial ice thickness and the target position uncertainty at 45 d lead time. The choice of that lead time is somewhat arbitrary, yet it can be motivated by two points. First, the wind field can be expected to be decorrelated from its initial state, so that the previously established significance of wind predictability is small (see Sect. 4.2) and other effects with longer persistence gain more importance. Nevertheless, some imprint of wind predictability is still inherited from early lead times (see Sect. 4.1.3). Second, the spatial patterns of target position uncertainty do not change significantly after four to five weeks, so that the fixed time $t = 45$ d can be considered representative for a time period of at least two weeks. It is also the latest lead time enabling a direct comparison of January and July initializations, due to the chosen thresholds in the trajectory filtering.

For July, there is virtually no significant correlation of initial ice thickness and uncertainty in all models; small patches with significant differences cover barely more than 5 % of the area, meaning that no field significance is given. A different picture emerges for the initializations in winter. For AWI-CM1 there are persistent regions of significant negative correlation in the





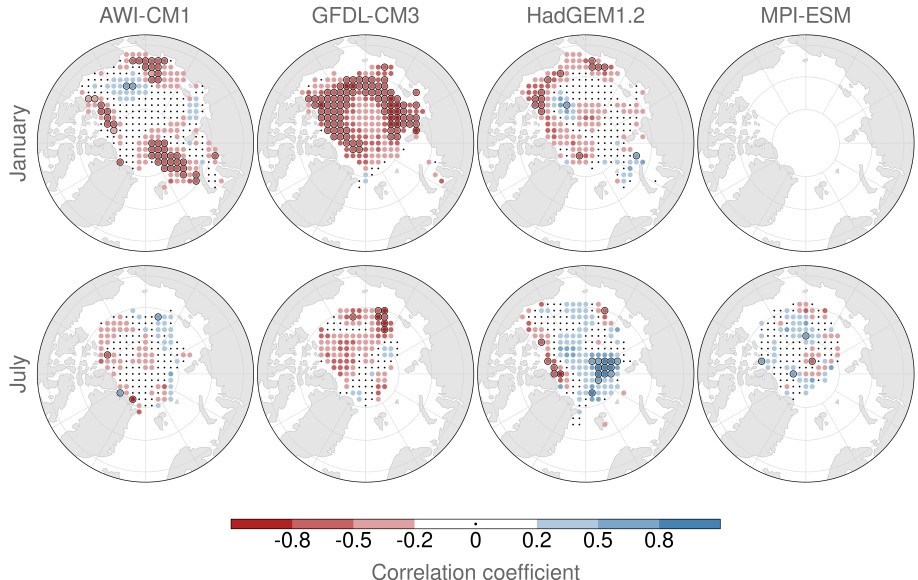

**Figure 10.** Correlation of target position normalized uncertainty at $t = 45$ d and initial sea ice thickness for each initial position for January (top) and July (bottom) initializations. Black dots indicate locations with correlation weaker than $\pm 0.02$. Colored dots encompassed by a black circle denote a $p$-value smaller than $0.05$. The different initializations, at least eight, serve as sample per target position.

East Siberian Sea, at the coast of the CAA and northeast of Svalbard. In these regions, a thinner initial ice cover yields a higher uncertainty (and thus lower predictability) of the target position prediction. In the open ocean there is no correlation of

uncertainty and ice thickness, as well as north of the coasts of Greenland, Ellesmere Island and Alaska. In HadGEM1.2 there is a ring-shaped pattern of weak to moderate negative correlation in the coastal regions and a few hundred kilometers towards the open ocean, albeit also rarely significant. While wind and ice drift uncertainty are in close correspondence for HadGEM1.2, ice thickness does not appear to play a significant role regarding ice drift uncertainty. Note that HadGEM1.2 simulates by far the thickest ice cover, so that negative thickness anomalies may still yield comparably thick ice. In GFDL-CM3, there is

generally a negative correlation of initial ice thickness and ice drift uncertainty. In large regions, this relation is significant, for instance in the coastal regions of the CAA, Greenland and Russia. There we observe a significant and strong relation of initial ice thickness, i.e. the thicker the ice at initial time, the less uncertain and thus more predictable is the target position prediction between $30$ and (at least) $45$ d lead time.

## 5   Discussion

While the temporal evolution of spatially averaged uncertainty is qualitatively similar across the models, the spatial characteristics, the quality of seasonal differences, and also the response to initial ice thickness are not. We attribute this to different representations of the forces acting on the ice, as well as different sea ice physics implementations. This makes it difficult to





generalize our results, but stresses the importance of using not one but a variety of models to study the predictability of the climate system.

The local correlation of Eulerian wind vector uncertainty to ice drift uncertainty in HadGEM1.2 and AWI-CM1, as well as the fact that ice drift is barely more predictable than wind, are strong hints for a dominant influence of atmospheric variability on ice drift predictability. Regarding our hypothesis of sea ice drift being less uncertain (that is, more predictable) than wind vectors, we find ambiguous results. Slightly enhanced predictability of ice drift versus wind is found for AWI-CM1 in both January and July up to at least two weeks lead time, but this is less clear for HadGEM1.2, raising the question of what drives

possible differences in the relation of ice drift and wind predictability between these two models. As presented in Sect. 3, the annual cycles of climatological monthly mean drift speed and wind speed differ strongly between AWI-CM1 and HadGEM1.2, and so does the ice thickness distribution. It appears plausible that these contrasts play a role in the model differences regarding ice drift versus wind predictability. However, albeit the perfect model approach is well suited to assess the climate system's inherent predictability, as done here, it does not readily reveal causal links and physical processes as sinks and sources of

predictability.

    It is unfortunate that no daily near-surface wind data is available for GFDL-CM3 and MPI-ESM. As a consequence, it is not possible to examine for example if the divergence of atmospheric states in MPI-ESM is delayed compared to the other models, which could explain the increased ice-drift predictability compared to the other models (see Figs. 6 and 7). This was not intended by the experimental design, where all models added uncorrelated white noise with equal amplitude to the sea-surface

temperatures as initial perturbations. In particular for MPI-ESM and AWI-CM1 one might expect that the perturbations affect the atmospheric states to diverge, on average, at the same time after initialization, given that they share the same atmospheric component. It is however possible that the different ocean resolution, numerical diffusion, or coupling scheme have an influence on how the perturbations evolve and affect the atmosphere. Wind data for GFDL-CM3 and MPI-ESM would also help to unravel some other differences in ice drift predictability between the models, such as the different spatial patterns and how they

depend on the season (Fig. 6).

    Another data-related caveat is that there is no daily (sub-)surface ocean velocity data available for the four examined models within the APPOSITE data set, so that we can not offer a view on the results in regard to the variability of ocean currents. Given that ocean drag is an important forcing term for sea ice motion on the time scales considered in this study, this should be examined in future studies.

One shortcoming of analyzing Lagrangian trajectories is the loss of virtual buoys close to the ice edge. Therefore, owing to the chosen trajectory selection, the variability of the marginal ice zone, one very dynamic part of the ice cover, can not be represented adequately.

    Furthermore, there are other choices for quantifying the uncertainty or (loss of) information contained in an ensemble prediction. We also performed large parts of the analyses using two other measures for uncertainty, namely the normalized root

mean squared error (NRMSE, see Collins (2002)) and also the square root of the ellipse area instead of the semi-major axis length. This did not alter the results significantly.





We suggest that future studies should quantify the predictability of specific (scalar) components of the ice drift separately. For example, the drift speed could be the aspect of ice drift that is the most sensitive to the initial ice thickness and could thus reveal such a dependence more readily. It would also be interesting to account for the predominant wind direction in coastal
areas to examine if ice strength (and thus thickness) plays a larger role in determining drift uncertainty when wind drag is directed towards the shore, compared to winds directed off-shore or parallel to the coast.

In addition to that, the significance of ice thickness anomalies for ice drift uncertainty can be scrutinized more thoroughly following the approach of Day et al. (2014), where the initial ice thickness was replaced by a climatology in additional perfect-model simulations, which were then compared to the simulations with thickness initialization. For any future model simulations
related to ice-drift predictability, we strongly recommend that daily near-surface wind and (sub-)surface ocean current data is provided to enable a detailed examination of the physical drivers of sea-ice drift.

## 6   Summary and conclusion

In this work, we determined and analyzed the uncertainty of initialized Arctic sea ice drift predictions in four global climate models. We made use of a set of perfect-model experiments carried out in the APPOSITE project and calculated trajectories of
virtual ice floes with a newly implemented open-source trajectory tool. Our study provides a missing piece in the predictability analysis carried out in the APPOSITE project.

For the Lagrangian target position, spatial gradients of the uncertainty develop within few days. Spatial patterns vary between the models and from summer to winter. The spatially averaged uncertainty for a common subset (across all four models) of initial positions does not reach the climatological saturation value for at least $90$ d for predictions initialized on 1st January,
and for at least $45$ d for predictions initialized on 1st July. There are regions where the uncertainty is significantly larger for July initializations, but these regions vary between the models. For local ice velocity at a fixed position (Eulerian view), the uncertainty saturates within the first three to four weeks lead time, and spatial patterns are more intermittent than for the target position (Lagrangian view). The uncertainty is on average higher in July, yet large variability renders this difference statistically insignificant for most target positions.
While the wind variability explains large fractions of the uncertainty of the ice velocity vector, the initial ice thickness as a proxy for ice strength and internal forces was found to play a statistically significant but quantitatively small role regarding the target position predictability. There remain open questions, particularly about the origin of the spatial patterns of uncertainty and the summer-to-winter variation and about the role of (sub-)surface ocean currents, that call for additional studies on the predictability of sea-ice drift.





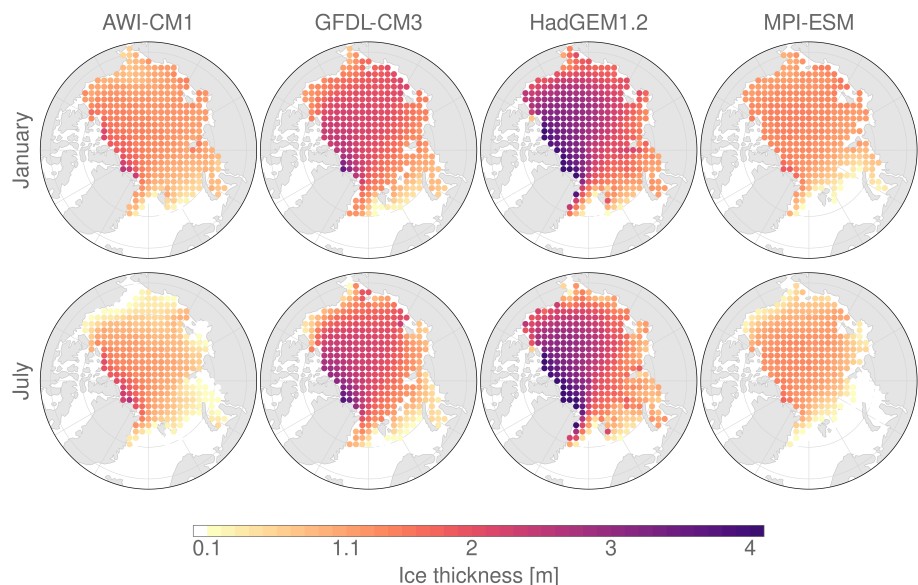

**Figure A1.** Climatological monthly mean sea ice thickness for January (top) and July (bottom), derived from the control run.

**Appendix A**





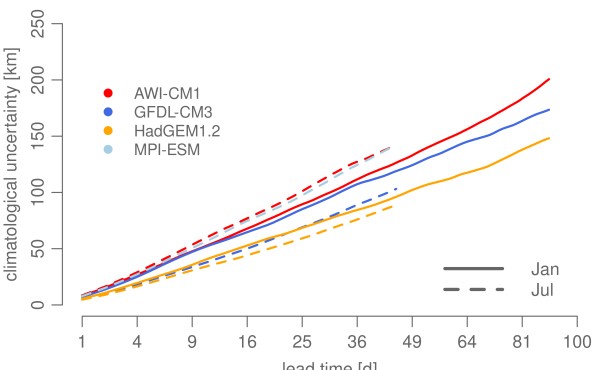

**Figure A2.** Temporal evolution of climatological uncertainty, i.e. the length of the semi-major axis of the variance ellipse. Note the square root horizontal axis.

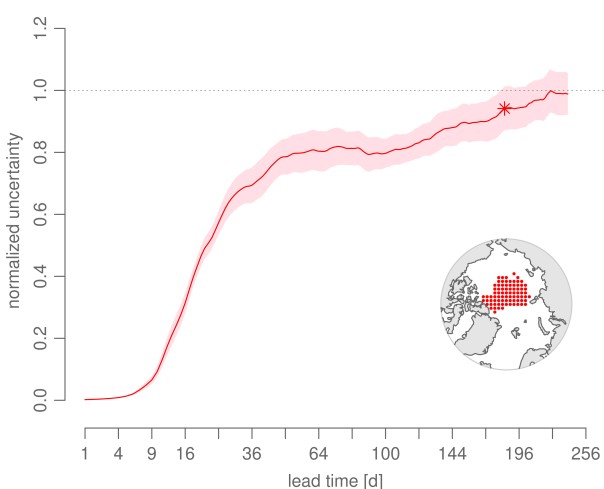

**Figure A3.** Additional simulation for AWI-CM1 with longer lead time. The time series shows the normalized uncertainty for the initial positions on the inset map. The asterisk marks the lead time when the climatological saturation value is attained for the first time (one-sided t-test, 0.95 level): there is no infinite predictability in Lagrangian perspective. Note the square root horizontal axis.

**Figure A4.** Maps of the correlation coefficient of the normalized uncertainties of Eulerian ice drift and near-surface wind vectors at $t = 12$ d lead time (colored dots), using the different initializations (at least eight) as sample. Locations with $p < 0.05$ are marked with a black ring, locations with a correlation weaker than $\pm0.2$ are denoted by a small black dot. There is a strong and significant positive correlation of sea ice drift uncertainty and near-surface wind vector uncertainty.



**Table A1.** Maximum and selected percentiles for sea ice thickness on the common subset of initial positions, derived from the control simulations.

| Model | max. [m] | 95% [m] | 50% [m] |
|---|---|---|---|
| *January* | | | |
| AWI-CM1 | 2.51 | 1.70 | 1.18 |
| GFDL-CM3 | 3.51 | 2.41 | 1.77 |
| HadGEM1.2 | 4.51 | 3.55 | 2.11 |
| MPI-ESM | 1.86 | 1.46 | 1.26 |
| *July* | | | |
| AWI-CM1 | 2.60 | 1.73 | 0.79 |
| GFDL-CM3 | 3.73 | 2.74 | 1.72 |
| HadGEM1.2 | 5.07 | 3.68 | 1.90 |
| MPI-ESM | 1.79 | 1.36 | 1.00 |

*Code and data availability.* APPOSITE data is available at http://data.ceda.ac.uk/badc/apposite/data. The developed trajectory software is available in the GitHub repository FESOM/spheRlab (commit fa31a05). Trajectory data are available from the authors under request.

*Author contributions.* H.G. and S.R. planned the research. S.R. implemented the trajectory tool and performed the APPOSITE data processing and the trajectory calculations. S.R. and H.G. analyzed, interpreted, and discussed the results. S.R. wrote the manuscript, with
contributions from H.G.

*Competing interests.* The authors declare that they have no conflict of interest.

*Acknowledgements.* This work was carried out as part of the Young Investigator Group Seamless Sea Ice Prediction (SSIP), funded by the Federal Ministry of Education and Research of Germany (grant: 01LN1701A). Data storage and computational resources were kindly provided by the German Climate Computing Center (DKRZ). We thank the coordinators of and contributors to the APPOSITE project for
making their data set available, and Ed Blanchard-Wrigglesworth for valuable discussions during the early stage of the project.



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
