# Peer review of "Predictability of Arctic Sea Ice Drift in Coupled Climate Models"

_The Cryosphere, 2022_

## Author Comment (AC1)

**Response to Anonymous Referee #1**

In the following, we present the referee comments in black, our point-by-point response and changes in the manuscript in blue, and literature references at the end of the document.

The manuscript "Predictability of Arctic Sea Ice Drift in Coupled Climate Models" discusses potential predictability of sea ice motion in four climate models from Eulerian and Lagrangian perspectives. The authors identify the potential predictability horizon and identify the wind variability as the main source of uncertainty. The role of initial ice thickness was found to be small. The manuscript is very well written, with good presentation of methodology and results, with deeply though discussions. It provides an important contribution to understanding of sea ice predictability in general. My minor comments only concern few overly complicated explanations in the text that should be simplified for less experience readers.

We thank Referee #1 for reviewing our work and for the valuable feedback and constructive comments, which helped to improve the quality of the manuscript, particularly clearing up overly technical explanations in the introduction and methods section.

**Line 30**: Although 'errors' and 'uncertainty' are well established terms, it is better to provide here a concise and clear definition of these terms (as well as 'accuracy' and 'skill') as understood by the authors for avoiding ambiguity in the rest of the manuscript.

We thank Referee #1 for this valuable suggestion. We changed the manuscript accordingly as follows. We replaced lines 30-38, that is,

"Initialized predictions inevitably come with errors and uncertainty. Errors arise from physical models being simplified representations of reality, incomplete knowledge of the initial conditions, and inevitable chaotic error growth (Lorenz, 1969, 1975), which gave rise to ensemble forecasting. Here, we therefore differentiate between *errors*, which one should strive to reduce, and *uncertainty*, which must be represented (and communicated) adequately. Both of them act on the forecast accuracy, or skill.

An initialized forecast is commonly considered skillful as long as its accuracy is higher than some chosen benchmark, for example a climatological reference forecast.",

with

"Initialized predictions inevitably come with errors and uncertainty. In this work, we differentiate between *errors*, that originate from *how* a forecast is made, e.g. from physical models being simplified representations of reality, incomplete knowledge of the initial conditions, or truncation errors in numerical models; and (inherent) *uncertainty* due to the inevitable growth of infinitesimal perturbations in the initial conditions, a property of the predicted (chaotic-deterministic) system (Lorenz, 1969, 1975). Errors should be reduced as much as possible, while uncertainty must be represented and communicated adequately. Both act on the forecast accuracy, or skill.

*Accuracy* refers to the "degree to which forecasts correspond to observations" (Murphy and Winkler, 1992), often described by the mean squared error of an ensemble with respect to a "true" value. An initialized forecast is commonly considered *skillful* as long as its accuracy is higher than some chosen benchmark accuracy, for example from a climatological reference forecast. Forecast skill, often expressed via "skill scores", can therefore be understood as a relative measure of accuracy.".

**Line 69**: What is "climatological uncertainty"? The following explanation "the uncertainty of an ensemble forecast constructed from independent years simulated by the same model with constant mean climate and variance" seems very short and hard to understand.

We agree that the sentence is hard to digest. We suggest simplifying the sentence as presented below – as it is still part of the introduction – and refer the reader to Section 2.3 ("Measures of predictability") and Hawkins et al. (2016) for a more detailed explanation of how a control simulation should be set up in general, and how it is set up in the used simulations.

We suggest replacing

"Analogous to the forecast skill horizon, here we call a variable (potentially) predictable up to a certain lead time as long as the uncertainty of an initialized ensemble forecast due to chaotic error growth is

smaller than the expected climatological uncertainty, that is, the uncertainty of an ensemble forecast constructed from independent years simulated by the same model with constant mean climate and variance."

by

"Analogous to the forecast skill horizon, here we call a variable (potentially) predictable up to a certain lead time if the uncertainty of an initialized ensemble is smaller than the expected climatological uncertainty. This climatological uncertainty, or variability, is usually derived from a control simulation with constant climate, see Sect. 2.1, Sect. 2.3, and Hawkins et al. (2016).".

Does it mean that a model is initialized at some point of time, then it is run for several years (and external forcing is the same every year), then an ensemble is constructed from individual years, then the uncertainty of a predictand in this ensemble is computed and used as a reference?

Yes, this is correct.

Is there a reference to justify building the "climatological uncertainty" this way?

We suggest Collins (2002) and Hawkins et al. (2016). The first study provides a general overview of the problem of initial value (climate) predictability and presents the normalized root mean squared error (NRMSE) as a metric for predictability which is quite often used in predictability studies; the second one provides an introduction on how model experiments for predictability studies should be designed.

So – yes, it is common to build the climatological uncertainty this way. For scalar quantities the climatological standard deviation is often used, or its climatological expectation value.

For how many years should the model run?

For the perfect model approach used in this study, the control simulations should be in equilibrium, which requires some spin-up time beforehand. Furthermore, the control run serves as a background climatology. For a statistically robust assessment of this climatology, the control run must be "sufficiently" long.

In our case, all models were run for at least 100 years for spin-up, and then at least for 200 more years as control simulation. A sample size of 200 years is comparably large, considering that statistics of the real climate mostly use 30 years. However, as per Referee #1's next question, the spin-up time was chosen too short for some of the participating models.

What if the model doesn't stabilize around a constant climate and the "climate uncertainty" continues to grow with the number of years?

This is a very important aspect that does not become clear in our manuscript yet, which is why we suggest the changes presented below.

Day et al. (2016) report that some of the participating models still exhibited linear trends in sea ice extent and volume, for instance. Thusly, the climate is only approximately constant, and these trends needed to be removed in their study, and related studies.

For sea ice drift speed however, we found linear trends of monthly mean ice speeds in January and July to be negligible, they were smaller than $3.1 \times 10^{-2}$ cm s$^{-1}$ decade$^{-1}$ for the examined models, and trends of the standard deviation of monthly mean ice speeds are in the same order of magnitude. We therefore did not remove any trend from either variable, as we expect the non-trivial removal of a trend in ice thickness, for instance, to complicate the analysis of drift speeds and ice velocity predictability in relation to the initial ice state later on.

We added in line 149 (Section 2.1, "APPOSITE data set"):

"However, Day et al. (2016) report that most participating models were not in equilibrium after the spin-up period; there was significant drift regarding sea ice extent and volume that needed to be removed prior to the analysis of predictability of these variables. We find trends in monthly mean ice drift speeds to be negligible (smaller than $3.1 \times 10^{-2}$ cm s$^{-1}$ decade$^{-1}$) so that no trends were removed in this study.".

We added in line 464 (Section 5, "Discussion"):

"While monthly mean ice speeds for January and July did not exhibit noteworthy linear trends, we again mention that the models were not in an equilibrium state after the spin-up period. This might have a more meaningful effect on possible future studies on the relation of ice speed (predictability) and the mean ice state.".

If my understanding is wrong, a better explanation, possibly with a scheme, is worth adding here. On such a scheme the error, uncertainty, accuracy and predictability can be visually shown for easier understanding by readers not well familiar with the topic.

From our perspective, the understanding of Referee #1 is very accurate. Still, we hope to have cleared up open questions by the changes in the manuscript and would refrain from adding a scheme if the editor and Referee #1 do not object.

**Line 208–211**: It is difficult to understand how the measure of uncertainty is computed. "variance ellipse", "semi-major axis" In which space? Dimensionality of this space? Can an equation be added here?

For the velocity covariance, the respective ellipse described by the covariance matrix is in $u$-$v$-space, and the semi-major axis of the ellipse has the dimension of a velocity (in m s$^{-1}$, for instance). For the position vectors, the measure has the dimension of a length (e.g. in km), and the variance is calculated in a local Cartesian coordinate system, obtained by a coordinate transformation of the geographical positions.

We agree that this should be presented more concisely by providing equations. We therefore added the following part in the Appendix of the manuscript and shortened lines 205-211 as per suggestion of Referee #2 from

"To account for the bivariate character of position and velocity vectors, we chose a different approach here, which we exemplify in the following for velocity vectors. For a given ensemble of velocity vectors at a given position and lead time, we determine the variance ellipse. Our measure for the uncertainty is then the length of the semi-major axis, which is the spectral norm of the covariance matrix of the velocity vectors. This also enables an analysis of the axis ratio and thus the anisotropy of the uncertainty. The uncertainty of initialized forecasts is then given by the mean of all available initializations (at least eight, due to the filtering)."

to

"To account for the bivariate nature of velocity vectors, we describe ensemble spread at a given lead time by the corresponding covariance matrix $\Sigma$. Our measure for uncertainty is then the spectral norm of $\Sigma$, which is also the length of the semi-major axis of the ellipse described by $\Sigma$ (see Appendix A). One can thus use $\Sigma$ for analyzing the anisotropy of uncertainty as well.".

We added in the Appendix:

"We estimate the uncertainty of the ensemble mean of Eulerian velocity vectors by the spectral norm, that is, the square root of the largest eigenvalue, of the covariance matrix in $u$-$v$-space. This is equivalent to the semi-major axis $a$ of the covariance ellipse and can be computed as follows.

Let $u_j$ and $v_j$ be the sea ice velocity components of the $j$th member of an ensemble of size $N_{\mathrm{mem}}$ at a fixed lead time, and $\bar{u}$, $\bar{v}$ be the respective ensemble means. The covariance matrix $\Sigma$ is given by

$$\Sigma = \begin{bmatrix} \sigma_u^2 & \sigma_{uv} \\ \sigma_{uv} & \sigma_v^2 \end{bmatrix}, \tag{1}$$

where $\sigma_u^2$ and $\sigma_v^2$ are the variance in the direction of $u$ and $v$, respectively, and $\sigma_{uv}$ the covariance:

$$\sigma_u^2 = \frac{1}{N_{\mathrm{mem}}} \sum_{j=1}^{N_{\mathrm{mem}}} (u_j - \bar{u})^2, \tag{2}$$

$$\sigma_v^2 = \frac{1}{N_{\mathrm{mem}}} \sum_{j=1}^{N_{\mathrm{mem}}} (v_j - \bar{v})^2, \tag{3}$$

$$\sigma_{uv} = \frac{1}{N_{\mathrm{mem}}} \sum_{j=1}^{N_{\mathrm{mem}}} (u_j - \bar{u})(v_j - \bar{v}). \tag{4}$$

The largest eigenvalue of $\Sigma$, i.e. the length of the semi-major axis $a$ of the variance ellipse, can then be obtained via

$$a^2 = \frac{1}{2}(\sigma_u^2 + \sigma_v^2) + \sqrt{\frac{1}{4}(\sigma_u^2 + \sigma_v^2)^2 - (\sigma_u \sigma_v - \sigma_{uv}^2)}, \tag{5}$$

$$b^2 = \frac{1}{2}(\sigma_u^2 + \sigma_v^2) - \sqrt{\frac{1}{4}(\sigma_u^2 + \sigma_v^2)^2 - (\sigma_u \sigma_v - \sigma_{uv}^2)}, \tag{6}$$

where $b$ is the length of the semi-minor axis for the sake of completeness. The value of $a$ represents the (direction of) maximum variability within the bivariate data, and we therefore consider it an appropriate measure for the uncertainty of the ensemble mean.

For the Lagrangian target positions we follow the same approach, except that we project the (spherical) geographical coordinates onto a (Cartesian) $x'$-$y'$-plane before as follows. Let $\lambda_j'$ and $\phi_j'$ be longitudes and latitudes from a trajectory ensemble at a fixed lead time in a rotated coordinate system, such that the North Pole of the rotated system represents the center of mass (barycenter) of the positions $[\lambda_j', \phi_j']$. The projection is then readily obtained by

$$x_j' = R \, \cos\phi_j' \, \cos\lambda_j', \tag{7}$$
$$y_j' = R \, \cos\phi_j' \, \sin\lambda_j', \tag{8}$$

with the Earth's radius $R = 6371$ km. Then one can plug in $x_j'$ and $y_j'$ for $u_j$ and $v_j$ in the framework above. Note that, due to the coordinate rotation, it holds $\bar{x}' = \bar{y}' = 0$.".

**Line 229-231**: This sentence is also difficult to digest. How a plane can be tangential to a point (barycenter)? Please add an equation.

We agree that this sentence needs to be revised. We therefore simplified the bulky version

"We follow the same approach for the position vectors, only that we first use an orthographic azimuthal projection onto the plane tangential to the barycenter of the point cloud given by a single ensemble prediction from a given initial position, for obtaining two-dimensional Cartesian coordinates (in km)"

to

"We follow the same approach for the position vectors, only that these vectors in geographical coordinates are projected onto a local Cartesian coordinate system (with units km) before, see Appendix A.",

and therewith refer to the equations added in response to the previous comment.

**Line 352**: What does it mean "normalized uncertainty reaches the climatological uncertainty"? Wasn't the normalization done to the climatological uncertainty? (eq. 4)? Shouldn't it read "uncertainty reaches the climatological uncertainty, i.e. normalized uncertainty reaches 1"?

We thank Referee #1 for this attentive remark. This is correct, the normalized uncertainty cannot reach the climatological uncertainty. We therefore changed

"...the normalized uncertainty reaches the climatological uncertainty ..."

simply to

"...the uncertainty reaches the climatological uncertainty ...".

**Line 410**: A reference to Fig. 10 should be added.

We added the missing reference in line 413, changing

"For each initial position, we calculate the correlation coefficient for initial ice thickness and the target position uncertainty at 45 d lead time."

to

"For each initial position, we calculate the correlation coefficient for initial ice thickness and the target position uncertainty at 45 d lead time (see Fig. 10).".

**References**

Collins, M. (2002). Climate predictability on interannual to decadal time scales: the initial value problem. *Climate Dynamics*, 19(8):671–692.

Day, J. J., Tietsche, S., Collins, M., Goessling, H. F., Guemas, V., Guilliory, A., Hurlin, W., Ishii, M., Keeley, S., Matei, D., Msadek, R., Sigmond, M., Tatebe, H., and Hawkins, E. (2016). The Arctic Predictability and Prediction on Seasonal-to-Interannual TimEscales (APPOSITE) data set version 1. *Geosci. Model Dev.*, 9:2255–2270.

Hawkins, E., Tietsche, S., Day, J. J., Melia, N., Haines, K., and Keeley, S. (2016). Aspects of designing and evaluating seasonal-to-interannual Arctic sea-ice prediction systems. *Quarterly Journal of the Royal Meteorological Society*, 142(695):672–683.

---

## Author Comment (AC2)

**Response to Anonymous Referee #2**

In the following, we present the referee comments in black, our point-by-point response and changes in the manuscript in blue, and literature references at the end of the document.

**Summary**:

This paper explores the limits of predictability of sea ice drift in four "perfect-model" simulations, and finds that the uncertainty in the winds is the primary limit to predictability. The thickness of the sea ice in one of the four models shows a negative correlation with position uncertainty. This is an interesting paper that should be accepted after mostly minor suggestions.

We thank Referee #2 for reviewing our work and for the valuable feedback and concise and constructive comments, which helped to improve the language and overall quality of the manuscript.

**Minor Comments:**

The ice speeds discussed in section 3, and shown in Figure 2 of 10 cm/s in the models seem really fast compared to observations which seem to be less than 5 cm/s. For example, https://nsidc.org/cryosphere /seaice/processes/circulation.html, shows that the typical ice speed less than 5 cm/s. And looking at some other recent papers such as Kwok, et al. 2013 (https://doi.org/10.1002/jgrc.20191) some similar numbers. Please discuss possible implications that the faster model speeds may have on the conclusions of this paper.

We thank Referee #2 for raising this concern regarding the role of ice speeds.

We acknowledge that especially the models MPI-ESM (12 cm s$^{-1}$) and AWI-CM1 (10 cm s$^{-1}$) exhibit annual mean ice speeds that are higher than common values in the literature, yet not unphysically high: Spreen et al. (2011) report a seasonal cycle of ice drift speed in the Arctic basin between approx. 6 and 12 cm s$^{-1}$ from satellite data, Zhang et al. (2012) obtain annual mean daily drift speeds around 7.5-8.5 cm s$^{-1}$ from buoy data after the year 2000, and Olason and Notz (2014) obtain a monthly climatology of 12-hourly mean drift speeds of about 7.9 cm s$^{-1}$ in January and 8.3 cm s$^{-1}$ in July, also from buoy data. GFDL-CM3 (annual mean 8 cm s$^{-1}$) and HadGEM1.2 (7 cm s$^{-1}$) lie thus within the observed range of observed ice speed distributions. However, we fully agree with Referee #2 that the role of model speeds is far from trivial regarding the conclusions of this manuscript. Therefore, we shall discuss this in the following.

If we were analyzing the performance of a model predicting "real-world" ice drift, for instance for operational ice drift forecasts, too high model speeds would result in a systematic forecast bias, certainly calling for action.

Here, as we "only" assess the growth of uncertainty due to the system's sensitivity to perturbations of the initial conditions, a direct comparison to observations is in a strict sense not necessary; within the respective model climate, the ice speeds are - semantically a bit of a stretch - "realistic" per the given assumptions of the perfect-model approach. Nevertheless, the results of our study can only have relevance for the real climate system if the models in use describe the real climate sufficiently well, particularly in terms of variability. In our case, the drift speeds are in fair agreement with observations, and the observed large-scale ice drift circulation patterns are reproduced by the models (not shown).

As Day et al. (2016) report as part of the description of the given simulations, mean and variability of the sea ice state differs considerably between the models, while each of the models has documented strengths and weaknesses in representing key features of Arctic climate. For instance, the APPOSITE simulations from MPI-ESM and AWI-CM1 systematically underestimate the monthly mean sea ice volume compared to observations. The faster drift for AWI-CM1 and MPI-ESM might be a physically sound consequence of the relatively low ice volume and thickness (i.e., ice being thinner and more mobile), albeit being a slightly less faithful representation of the mean state of the real climate. Also note that the simulations use a fixed present-day radiative forcing. This may lead to an equilibrium state with higher ice speeds than from the real transient climate of the past few decades.

That said – how does this model diversity with respect to ice speed impact our conclusions?

The uncertainty of an initialized prediction of a target position for a given model will likely grow faster if the ice moves faster (in this model), as the trajectories diverge more quickly. This also holds for

the climatological reference ensembles, which we use for normalization. Therefore, the effect of higher drift speeds on our results is largely compensated by the normalization. This is not to say there is no effect on the uncertainty of the initialized forecast, it just does not imprint much on the normalized uncertainty, which might also explain the following: Arguably our main result is the wind uncertainty being the limiting factor of ice drift predictability, and the observed close correspondence held for both the model with the lowest annual mean drift speed (HadGEM1.2) and AWI-CM1 with relatively fast drift, while both models also differed strongly with respect to their mean sea ice state, e.g. ice thickness and volume.

Considering the compensating nature of the normalization and the fact that perfect model simulations are not necessarily (designed to be) accurate predictions for the real climate, we argue that the main conclusions of our work remain valid, and at the same time we suggest that the role of ice speed (and drift direction) should receive more attention in future studies of ice drift "perfect-model" predictability.

We added the following sentence in Section 3 for clarification:

New in line 241: "Thus, the models also differ in how well they capture the current climate of the real system. Albeit the assessment of inherent predictability of the climate within a given model does not build upon the degree of accuracy to which it reproduces the real system, it is worth noting that each of the coupled general circulation models has individual strengths and shortcomings, particularly as predictability may depend on the mean model state."

To fix a then broken reference, we changed "This order" into "The aforementioned order" in line 242.

**Line 1**: I think it is worth restating Nansen's rule of thumb explicitly here.

We agree with this suggestion and changed the first sentence

"More than 120 years have passed between Nansen's empirical "rule of thumb" for sea ice drift (Nansen, 1902) and the latest developments of today's sophisticated dynamic sea ice modeling systems."

into

"More than 120 years have passed between Nansen's empirical "rule of thumb" about sea ice drifting 20° to 40° to the right of the wind direction at about 2 % of the wind speed (Nansen, 1902) and the latest developments of today's sophisticated dynamic sea ice modeling systems.".

**Line 32**: The paper tends to be too wordy. This and other comments below are aimed at tightening up the text. For example, on line 32, the authors write "Here, we therefore differentiate...". I would tighten this up to simply state "We differentiate...". I would comb through the paper and reduce the use of these transition words.

We thank Referee #2 for this feedback. We revised line 32 accordingly and follow the related suggestions in the other comments.

**Line 72-93**: Lines 72 -93 seemed out of place. I would maybe move it up above line 58? I don't feel strongly about this.

If Referee #2 and the editor do not object, we suggest keeping the current structure, separating the studies regarding forecast skill (in the real system) presented in lines 43–57 from the studies on inherent predictability (in the perfect-model world) in lines 75–92.

**Line 73**: site a few "recent studies".

We recognize that our phrasing was ambiguous. The studies we intended to refer to are presented in lines 75 to 88. We therefore changed

"...has been assessed in a number of recent studies. The following studies are all ..."

into

"...has been assessed in several recent studies, presented in the following. These are all ...".

**Line 81**: Line 81 stating "(two-dimensional)" is not necessary since this should be implied by the

discussion of area.

Revised accordingly, deleted "(two-dimensional)".

**Line 154**: I suggest stating the "1st July" as "1 July" or "July 1st" or "the 1st of July"., then restate "1st January" in the same way.

We thank Referee #2 for this suggestion and restated all occurrences as "1 July" and "1 January", that is, in lines 154, 181, 338, 489, 490, and the caption of Figure 6.

**Line 202**: delete "which are both two-dimensional quantities".

Revised accordingly.

**Line 205-210**: Too wordy. I think the authors can delete most of lines 206-207, and just go with lines 208-209.

As Referee #1 suggested adding more information on the computation of the uncertainty of an ensemble forecast, we added several equations in the Appendix and shortened lines 205-211 from

"To account for the bivariate character of position and velocity vectors, we chose a different approach here, which we exemplify in the following for velocity vectors. For a given ensemble of velocity vectors at a given position and lead time, we determine the variance ellipse. Our measure for the uncertainty is then the length of the semi-major axis, which is the spectral norm of the covariance matrix of the velocity vectors. This also enables an analysis of the axis ratio and thus the anisotropy of the uncertainty. The uncertainty of initialized forecasts is then given by the mean of all available initializations (at least eight, due to the filtering)."

to

"To account for the bivariate nature of velocity vectors, we describe ensemble spread at a given lead time by the corresponding covariance matrix $\Sigma$. Our measure for uncertainty is then the spectral norm of $\Sigma$, which is also the length of the semi-major axis of the ellipse described by $\Sigma$ (see Appendix A). One can thus use $\Sigma$ for analyzing the anisotropy of uncertainty as well.".

**Line 237-238**: Too wordy. I would delete the first sentence starting at line 237, and simply say "Maps of average ice thickness for the months of March and September are presented in Day et al. (2016)."

Revised accordingly.

**Line 246**: delete "previously introduced".

Revised accordingly.

**Line 274**: delete "In the following", and start the sentence as "We now consider the differences in the trajectories...".

Revised accordingly.

**Figure 4 and 5**: Combine Figures 4 and 5.

We recognize that this would group related information more effectively than in the given separated figures. In fact, in an earlier version of the manuscript, we combined Figures 4 and 5; trying out the two options of putting them next to each other and on top of each other. However, the size limit for figures (given by the printable area on the page) rendered the ellipses and displacement vectors hard to recognize, as the combined figure must be scaled down in both cases. If the editor and Referee #2 do not feel strongly about this, we suggest keeping Figures 4 and 5 separated.

**293**: delete "In the following, "

Revised accordingly.

**Line 294-295**: change "a (normalized) uncertainty" to "an uncertainty".

Revised accordingly.

**Figure 6**: Capitalize "Uncertainty" under colorbar.

Revised accordingly.

**Line 306**: delete "also".

Revised accordingly.

**Line 309**: change "with January and July initializations" to "for January and July".

Revised accordingly.

**Line 334**: delete "an additional point of view –".

We revised the sentence from

"(i) an additional point of view - the Eulerian perspective -, (ii) . . . "

into

"(i) an analysis from the Eulerian perspective, (ii) . . . ".

**Line 341**: delete this sentence

Revised accordingly.

**Line 344**: change ". . . position here. This enables. . . " to ". . . which enables. . . ".

Revised accordingly.

**Line 346**: change "of normalized" to "for"

Revised accordingly.

**Line 463**: delete "also".

Revised accordingly.

**Line 455**: change "affect" to "cause".

Revised accordingly.

**Line 485-486**: delete sentence starting with "Our study. . . ".

Revised accordingly.

**Line 487**: change "within few days" to "within a few days".

Revised accordingly.

**References**

Day, J. J., Tietsche, S., Collins, M., Goessling, H. F., Guemas, V., Guilliory, A., Hurlin, W., Ishii, M., Keeley, S., Matei, D., Msadek, R., Sigmond, M., Tatebe, H., and Hawkins, E. (2016). The Arctic Predictability and Prediction on Seasonal-to-Interannual TimEscales (APPOSITE) data set version 1. *Geosci. Model Dev.*, 9:2255–2270.

Olason, E. and Notz, D. (2014). Drivers of variability in arctic sea-ice drift speed. *Journal of Geophysical Research: Oceans*, 119(9):5755–5775.

Spreen, G., Kwok, R., and Menemenlis, D. (2011). Trends in arctic sea ice drift and role of wind forcing: 1992–2009. *Geophysical Research Letters*, 38(19).

Zhang, J., Lindsay, R., Schweiger, A., and Rigor, I. (2012). Recent changes in the dynamic properties of declining Arctic sea ice: A model study. *Geophysical Research Letters*, 39:20503–.